# Identifying Microbiome Dynamics in Pediatric IBD: More than a Family Matter

**DOI:** 10.3390/biomedicines11071979

**Published:** 2023-07-13

**Authors:** Nikolas Dovrolis, Anastasia Moschoviti, Smaragdi Fessatou, George Karamanolis, George Kolios, Maria Gazouli

**Affiliations:** 1Laboratory of Pharmacology, Department of Medicine, Democritus University of Thrace, 68100 Alexandroupolis, Greece; ndovroli@med.duth.gr (N.D.); gkolios@med.duth.gr (G.K.); 2Individualised Medicine & Pharmacological Research Solutions Center (IMPReS), 68100 Alexandroupolis, Greece; 3Third Department of Pediatrics, “Attikon” General University Hospital, Medical School, National and Kapodistrian University of Athens, 12462 Haidari, Greece; moschoviti.anastasia@gmail.com (A.M.); sfessatou@gmail.com (S.F.); 4Gastroenterology Unit, Second Department of Surgery, Aretaieio Hospital, Medical School, National and Kapodistrian University of Athens, 11527 Athens, Greece; georgekaramanolis@yahoo.co.uk; 5Laboratory of Biology, Department of Basic Medical Sciences, Medical School, National and Kapodistrian University of Athens, 11527 Athens, Greece; 6School of Science and Technology, Hellenic Open University, 26335 Patra, Greece

**Keywords:** Crohn’s disease, ulcerative colitis, microbiome, inflammatory bowel disease, family

## Abstract

Background: Pediatric inflammatory bowel disease (IBD) is a chronic inflammatory intestinal disease that affects both children and adolescents. Symptoms can significantly affect a child’s growth, development, and quality of life, making early diagnosis and effective management crucial. This study focuses on treatment-naïve pediatric IBD patients and their immediate families to identify the role of the microbiome in disease onset. Methods: Nine families with pediatric IBD were recruited, comprising seven drug-naïve Crohn’s disease (CD) patients and two drug-naïve ulcerative colitis (UC) patients, as well as twenty-four healthy siblings/parents. Fecal samples were collected for 16S ribosomal RNA gene sequencing and bioinformatics analysis. Results: We identified patterns of dysbiosis and hallmark microbial taxa among patients who shared ethnic, habitual, and dietary traits with themselves and their families. In addition, we examined the impact of the disease on specific microbial taxa and how these could serve as potential biomarkers for early detection. Conclusions: Our results suggest a potential role of maternal factors in the establishment and modulation of the early life microbiome, consistent with the current literature, which may have implications for understanding the etiology and progression of IBD.

## 1. Introduction

Pediatric IBD is a chronic inflammatory bowel disease that affects children and adolescents. The two main types of IBD are Crohn’s disease (CD) and ulcerative colitis (UC), which are characterized by inflammation and damage to the digestive tract. Children with IBD may experience a variety of symptoms, including abdominal pain, diarrhea, rectal bleeding, weight loss, and fatigue [1]. These symptoms can significantly impact a child’s growth, development, and quality of life, making early diagnosis and effective management crucial. Diagnosis of pediatric IBD can be challenging, as symptoms can be similar to those of other digestive disorders or infections. Pediatric gastroenterologists will often perform a variety of tests, including blood work, stool samples, endoscopy, and imaging studies, to help confirm a diagnosis [2]. Once a diagnosis has been made, treatment will typically involve a combination of medications and dietary changes [3,4]. Some children may require surgery in more severe cases [5]. One of the unique challenges of managing pediatric IBD is ensuring that treatment plans are appropriate for a child’s age and developmental stage. Medications may need to be adjusted based on weight and age, and some medications may not be appropriate for use in young children. In addition, dietary changes may be necessary to help manage symptoms, but it is important to ensure that a child’s nutritional needs are being met [4]. In addition to physical symptoms, pediatric IBD can also impact a child’s emotional well-being. Children with IBD may experience anxiety, depression, and social isolation, especially if they are unable to participate in normal childhood activities [6]. Overall, managing pediatric IBD requires a multidisciplinary approach, involving gastroenterologists, dietitians, mental health professionals, and other healthcare providers. Regular monitoring and follow-up appointments are important to ensure that symptoms are managed effectively and to identify any potential complications. With proper diagnosis, treatment, and support, children with IBD can lead healthy and fulfilling lives.

Although the cause of IBD is not fully understood, various potential factors have been linked to its development and progression. These include gene expression dysregulation and polymorphisms [7,8], lifestyle choices, and the role of microbiota [9,10]. Despite research, it remains unclear why the disease affects people of all ages. Previous studies have suggested that genetics play an important role in developing IBD [11,12,13] with an up to 12% increased risk for disease occurrence [14]. More recently, with the rise of microbiome-focused research, it has been shown that the intrafamilial microbiome can be linked to specific genetic traits and be studied for its behavior during dysbiosis [15,16]. It has also been reported that microbiota composition is shared between cohabitating individuals [17,18,19] but also displays patterns of similarity that differ between families [20]. Perhaps the most important questions shared by scientists today revolve around whether the microbiome can act as a mediator, a causative agent of disease, or is just its innocent victim [21,22,23,24,25].

Microbiome analysis via bioinformatics (metagenomics analysis) is a rapidly growing field that focuses on analyzing the genetic material of microbial communities present in different environments, including the human body [26]. This involves the use of computational tools and statistical methods to study the diversity and abundance of microorganisms and their functional characteristics [27]. With the advent of high-throughput sequencing technologies, it has become possible to study the microbiome at an unprecedented level of detail, enabling researchers to explore the relationship between microbial communities and human health and disease [28,29]. These techniques play a crucial role in identifying specific microbial taxa associated with different diseases and understanding their functional roles. It also aids in the development of diagnostic and therapeutic tools that can modulate the microbiome to treat various diseases. Moreover, metagenomics analysis is essential for interpreting the vast amount of data generated by microbiome studies, allowing researchers to identify patterns and relationships that are otherwise difficult to detect. As the field continues to expand, it is likely to lead to new insights into the complex interplay between microbial communities and their host organisms, paving the way for the development of personalized medicine approaches that consider an individual’s unique microbiome composition.

In this spirit, our study explores the dynamics of the microbiome within families of pediatric IBD patients, attempting to understand its characteristics during dysbiosis and the different microbial profiles formed within a family. Our methodology was structured for the identification of dysbiosis patterns and hallmark microbial taxa among these patients and their healthy first-degree relatives (parents and siblings). This allowed us not only to present the pediatric microbiome in a unique way but also to identify potential markers of diagnosis and treatability. 

## 2. Results

### 2.1. IBD versus HC Overall

To identify differences between IBD patients and healthy controls (HC), several analyses were performed for evaluating the quantitative and qualitative differences between them. This allows us to study the potential of microbial taxa as biomarkers for IBD in general and also compare our results to previous works. 

#### 2.1.1. Microbiota Composition and Univariate Analysis

The overall relative abundance of microbial phyla when comparing HC and IBD samples presents an increase in *Bacteroidetes* (10% increase) and *Proteobacteria* (7% increase) in patients while *Firmicutes* are decreased (16% decrease) (Figure 1A). On the genus taxonomic level in relative abundance, the patient samples are characterized by an increase in *Bacteroides* (17%) and a decrease in *Faecalibacterium* (7%), among other changes (Figure 1B). The univariate analysis of phyla provides statistical significance (*p.adjust <* 0.05) for *Bacteroidetes* and *Proteobacteria* only when the genera *Veillonella*, *Haemophilus*, *Granulicatella*, *Erysipelatoclostridium*, *Shigella*, and *Streptococcus* are significantly increased and members of *Candidatus Soleaferrea* are decreased. In addition, the species *Veillonella parvula*, *Streptococcus parasanguinis*, *Haemophilus parainfluenzae*, *Granulicatella paradiacens*, *Ruminococcus flavefaciens*, *Dorea massiliensis*, *Shigella sonnei*, *Bacteroides fragilis*, *Bacteroides acidofaciens*, and *Bacteroides caccae* are significantly increased (FC > 8 and *p.adjust* < 0.05) while *Ruminococcus flavefaciens* and *Alistipes massiliensis* are significantly decreased (FC > 6 and *p.adjust* < 0.05). The top 10 over- and under-abundant species and genera in the patient group versus all healthy controls are depicted in Table 1. All univariate analysis results for the aforementioned taxonomic levels via DESEQ2 can be found in Appendix A.

#### 2.1.2. Diversity Metrics

When comparing the pool of the nine pediatric IBD patients against the pool of healthy family members (Figure 2A), alpha diversity appears to be reduced, but without achieving statistical significance between the two groups (Kruskal–Wallis *p* = 0.11). The same effects are also true when subgrouping the HC group according to their familial relationship with the patients (Figure 2B), where the mother/father/brother/sister subgroups appear to exhibit comparable alpha diversities. The loss of statistical significance can be attributed to the wide distribution of values in the patient group, which will be discussed later on. The NMDS plot (Figure 3), used to depict beta diversity, presents clear clustering of the HC samples, while the patient samples are distributed differently. ANOSIM statistics of R = 0.19878 (*p* < 0.04) signify a high similarity between the groups.

### 2.2. Intrafamilial Microbiome Changes

By analyzing each family individually, we were able to identify patterns of microbiota behavior during IBD in our pediatric patients and ascertain that the microbiome is heavily affected by the disease in individuals who share lifestyles with our healthy controls. 

#### 2.2.1. Biodiversity

Comparing how diverse the microbiomes of family members are shows that pediatric IBD patients diverge significantly. In Figure 4, all a-diversity indices of members of the nine families are shown individually. In six of the nine families, the patients exhibit less biodiversity compared to their healthy family members, while in the rest, they appear to be richer in microbiota from their parents/siblings. This finding, although surprising, can be explained when comparing the alpha diversity indices of patients only, which shows that they are more or less equal, with the exception of Patients 4 and 5, for whom the alpha diversity indices were further decreased. For example, Patients 2 and 9 appear to have lower and higher, respectively, Shannon indices in Figure 4, but the comparison in Figure 2C shows that there is no discernable difference between them. In addition, we observed no differences between children with CD and UC.

#### 2.2.2. Similarity Clustering

As part of our investigation into the dynamics of the microbiome within families of pediatric IBD patients, we utilized Ward’s hierarchical clustering to identify closer similarities among the microbiomes of specific family members. Figure 5 displays the resulting dendrograms, which revealed that in most cases, the microbiome of the patient was found to be either closer to that of the mother or diverging from all other family member microbiomes. 

#### 2.2.3. Microbial Composition

To extend our previous analyses of microbiota differential abundance, we performed the same analysis in each family, comparing the pediatric patients against a pool of their family members. Additionally, we calculated the intersection of microbial genera (Figure 6A) and species (Figure 6B) with increases and decreases in their counts. This analysis enabled us to identify patients who exhibit shared differentially abundant taxa. For instance, patients belonging to Families 1 and 2 have only two enriched genera in common (*Butyricimonas* and *Lachnoclostridium*) but share 12 genera (*Anaerosporobacter*, *Anaerobacterium*, *Prevotella*, *Dorea*, *Holdemanella*, *Paraprevotella*, *Clostridium*, *Ruminococcus*, *Lachnobacterium*, *Pantoea*, *Peptococcus*, and *Shigella*) that demonstrate decreased population levels. Similarly, Patients 3 and 6 share six enriched genera (*Veillonella*, *Haemophilus*, *Shigella*, *Pantoea*, *Bifidobacterium*, and *Flavonifractor*) and eighteen genera (*Phascolarctobacterium*, *Akkermansia*, *Anaerobacterium*, *Trigonala*, *Holdemanella*, *Intestinimonas*, *Spiroplasma*, *Alistipes*, *Caloramator*, *Clostridium*, *Ruminiclostridium*, *Sporobacter*, *Vallitalea*, *Victivallis*, *Holdemania*, *Odoribacter*, *Coprobacter*, and *Porphyromonas*) displaying reductions in population size. Because of the paucity of samples in each family, we disregarded statistical significance and considered all taxa with a fold change in abundance greater than two (FC > 2). The genera *Butyricimonas* and *Veillonella* exhibited increased abundance in the patients of at least six families (considering all multiple combinations), while *Anaerosporobacter*, *Phascolarctobacterium*, *Akkermansia*, *Anaerobacterium*, *Anaerovorax*, *Lachnospira*, *Prevotella*, and *Trigonala* were decreased. In addition, the species *Butyricimonas virosa* and *Bacteroides xylanisolvens* were more abundant in the patients of at least six families (considering all multiple combinations), while *Lachnoclostridium xylanolyticum*, *Ruminococcus albus*, *Eubacterium eligens*, *Ruminococcus lactaris*, *Blautia luti*, *Anaerosporobacter mobilis*, *Holdemanella biforme*, *Ruminococcus bromii*, *Anaerobacterium chartisolvens*, *Trigonala elaeagnus*, *Coprococcus eutactus*, *Ruminococcus flavefaciens*, *Dorea formicigenerans*, *Akkermansia muciniphila*, *Anaerovorax odorimutans*, *Lachnospira pectinoschiza*, *Lactobacillus rogosae*, *Streptococcus salivarius*, and *Sporobacter termitidis* showed decreased numbers when compared with the healthy family members. Appendix A includes the comparative analysis of differential abundance between patients and their healthy relatives within each family. It also provides information on the shared genera and species that are either overabundant or underabundant across multiple families as those appear in Figure 6.

## 3. Discussion

This work focuses on treatment-naïve pediatric IBD patients and their immediate families to identify the role of the microbiome in disease onset. Our results elucidate the fact that even though family members share lifestyle and dietary habits, the patients exhibit unique microbiome patterns which could only be attributed to the disease. IBD-related dysbiosis shown here carries common characteristics among patients who share ethnic, habitual, and dietary traits among themselves and their families. In addition, we further examine the disease’s impact on specific microbial taxa and how those can serve as potential biomarkers for early detection. 

In terms of comparing IBD patients with healthy controls, our study successfully replicates and validates the findings of previous research, such as the loss of *Firmicutes* [30,31] and the increase in the phylum *Bacteroidetes* and the genus *Bacteroides* [32,33] along with *Proteobacteria* [34], which are well documented in IBD. In addition, several studies have shown that *Faecalibacterium prausnitzii*, which exhibits anti-inflammatory potential [35], is decreased significantly in IBD [36], a fact that can only partially be confirmed by this study since the genus *Faecalibacterium* shows a statistically significant reduction in patients but the specific species fails to achieve the required *p*-value cutoff of reporting. This can be attributed to the wide distribution of *F. prausnitzii* among all samples due to its universal presence but also given the fact that there are studies showing an actual increase in some pediatric patients [37], balancing out its statistical power. We also report on several other bacterial taxa differentially abundant between patients and healthy controls able to serve as potential IBD biomarkers as we have in previous works with disease activity [38] and response to treatment [39]. Our findings regarding *Prevotella* align with previous research [40,41], which observed a significant decrease or even depletion of this genus in IBD patients. In a systematic review of gut microbiota profiles in pediatric IBD patients [42], the genus *Lachnospira* was highlighted as being consistently underabundant in several studies [16,43,44,45,46], in agreement with our own findings. *Lachnospira* along with *Prevotella* are two well-established short-chain fatty acid (SCFA)-producing bacterial genera, and their paucity in IBD, and especially CD, is a known factor in promoting inflammation [47]. The role of short-chain fatty acids (SCFAs) in maintaining homeostasis and supporting overall health has been extensively discussed, particularly in relation to dietary habits, which have a significant impact on the composition of the microbiota [48]. Interestingly, despite the influence of geopolitical designations, religion, and daily life on dietary habits, a study conducted on new-onset pediatric Crohn’s patients from Saudi Arabia [49] reported findings consistent with ours regarding *Holdemanella* and *Lactobacillus*. Additionally, Sila et al. [50] reported a similar underabundance of *Lactobacillus* in newly diagnosed pediatric patients with IBD. Furthermore, a study by Malham et al. [51], investigating the role of the microbiome in predicting disease severity and diagnosis in pediatric IBD patients, revealed similar findings to our research regarding the underabundance of *Catenibacterium* and *Akkermansia* in patients. Notably, *Akkermansia* has also been reported to exhibit lower abundance in patients from at least two other studies [52,53]. Another interesting result was the overabundance of *Alcaligenes*, which, in a previous study [54], the authors had shown that in innate lymphoid-cell-depleted mice, the presence of *Alcaligenes* was sufficient to promote systemic inflammation, while systemic immune responses to it have been associated with Crohn’s disease. It is not uncommon for studies conducted in different countries, which inherently involve variations in microbiome profiles, to report on the same bacterial taxa with differences in terms of changes in abundance for a given condition. Streptococcus serves as an example, as it has been found to be overabundant in our pediatric IBD patients, consistent with findings from Kowalska-Duplaga et al. [43], Tang et al. [44], Wang et al. [55], Ijaz et al. [45], and Lewis et al. [56], while El Mouzan et al. [49] and Assa et al. [57] report an underabundance in pediatric IBD.

Considering the low biodiversity exhibited by patients in most families when compared to their healthy first-degree relatives, a few things can be deduced. Starting from the fact that even though the patients live in a microbiota-rich environment, they appear to be “resistant” to becoming hosts of more bacteria, and bacteria that attempt to colonize them are either instantly killed by an overactive immune system or do not find enough space to thrive. The spread of pathogenic taxa must either create a hostile microenvironment via metabolism or absorb all the nutrients which are crucial for the survival of the commensal ones. These hypotheses are in line with our current knowledge of dysbiosis [58,59]. Regarding the three families in which healthy relatives show reduced biodiversity when compared to patients, there might be a few reasons to explain the phenomenon, like pathogenic bacteria invasion in patients, strong but low diversity colonies in healthy controls which regulate their immunities, and differences in socioeconomic/dietary/environmental factors when compared to the other families. The fact that alpha diversity among patients is comparable in most cases hints at immunity mechanisms targeting taxa of specific functions, which needs to be further investigated since we were not able to find a specific cause for this when comparing patient samples according to their metadata. 

Clustering microbiome profiles according to their similarities has provided evidence that in most cases, pediatric patient microflora either resembles that of the mother or differs significantly from those of all healthy family members. This suggests a potential role for maternal factors in the establishment and modulation of the microbiome in early life, which is consistent with the literature and may have implications for understanding the etiology and progression of IBD. Previous works have shown that the human microbiome is shaped early in life (first 2–3 years) [60,61] and is modulated by maternal-related factors like mode of birth [62,63] and breastfeeding [64,65]. In our results, we could not identify these factors as co-factors of differentiation, but the close resemblance between the patient and maternal microbiome profiles is always an interesting finding, especially when we expand studies beyond neonates, infants, and children into adolescence. This is important because even though the maternal microbiome is essential in early life, adolescents (ages 10–19), due to school and social lives, often diverge from the pure familial environment, with studies expanding beyond common pathophysiological changes in known microbiota-associated disorders like IBD [66,67]. 

Another consideration for future works is how the non-IBD siblings’ microbiome might show signs of divergence from the parents’ “healthy plane”, as defined by Jacobs et al. [16]. For example, in our study, the siblings’ microbiomes are mostly clustered closer to those of the parents but show, in total, a slight drop in alpha diversity, as shown in Figure 2B. Additionally, some of the differentially abundant taxa characterizing patients observed here might also be prominent in those children. For example, since *Butyricimonas* and *Veillonella* appear to be prominent in the patients of most families, along with their increased abundance noted in the literature, they have the potential to serve as prognostic markers of disease occurrence. In Wang et al. [55], *Veillonella* not only demonstrates increased prevalence in patients undergoing anti-TNF treatment, but it also maintains a notable abundance even after therapy, suggesting resistance to regulation by external factors. In addition, *Veillonella* appears to be more abundant in several pediatric Crohn’s studies, as reported by a recent meta-analysis [42], suggesting a constant factor among diverse populations. Furthermore, *Butyricimonas*, with its elevated prevalence, can reveal immunological dysfunctions and impacts, as highlighted by Chen et al. [68], whose research suggests that adhesive bacteria, such as *Butyricimonas* found in the terminal ileum of pediatric patients, contribute to heightened activation of Th17 cells and the secretion of immunoglobulin A in the gut lumen. These mechanisms can contribute to the inflammatory processes, and given *Butyricimonas’* proximity to the intestinal wall, it represents a promising candidate for further investigation. Interestingly, butyrate-producing *Clostridia*, like *Erysipelatoclostridium* and *Butyricimonas*, were found in our study to be enriched in pediatric IBD patients, but not in other studies. This observation is intriguing, as butyrate production has been correlated with the mitigation of inflammation [69]. These discrepancies may be even more pronounced within other families, underscoring the importance of vigilant monitoring in children who may be susceptible to developing IBD. 

As is true for all works based on 16S rRNA amplicon sequencing, this study has the potential for small sample sizes to affect the accuracy and reliability of the results. This is because the technique relies on amplifying and sequencing a specific region of the 16S rRNA gene, which may not represent the entire microbial community present in a given sample (in our case the V3/V4 hypervariable regions). Additionally, small sample sizes can also increase the likelihood of sampling bias, as there may be significant variation in microbial communities between individual samples [70]. Furthermore, 16S rRNA amplicon sequencing is limited in its ability to accurately identify certain microbial species and may provide only a broad taxonomic classification of the microorganisms present. This can limit the depth of analysis and hinder the ability to detect important microbial interactions [71]. We have tried to alleviate this by utilizing a pipeline with strict taxonomic identification parameters and cross-checking the results with the literature, but uncertainty still exists when it comes to species-level analyses. Finally, the use of fecal samples provides more diluted information about IBD-related dysbiosis when compared to biopsies [72]. 

By delving into the intrafamilial microbiome, our investigation yields additional evidence that underscores the complex dynamics involved in the development and progression of IBD, particularly concerning treatment approaches. The modulation of some of the taxa highlighted here has been investigated as a potential therapeutic strategy, and our findings may provide more targets. *Candidatus Soleaferrea*, shown in our study to have reduced abundance in pediatric patients, has been found to be upregulated in Svolos et al. [73] after Exclusive Enteral Nutrition (ENT) and a food-based diet that simulates it. ENT is considered a first-line treatment for CD and its sub-phenotypes [74]. *Haemophilus* and the specific species *Haemophilus parainfluenzae* highlighted in this study are also promising candidates for modulation-based therapies since they were also highlighted as some of the taxa more abundant in first-diagnosis pediatric CD patients by Kansal et al. [75]. *Haemophilus parainfluenzae* is an opportunistic pathogen that can also be detected in the oral cavity and respiratory tract which a recent study [76] has linked to CD progression and severity. In a recent study by our group [77], we reported on a reverse correlation between *Granulicatella* and IFN type II expression levels, which holds promise as a prognostic marker for anti-TNF treatment. Similarly, modulating this taxon may have the potential to not only influence the occurrence and outcomes of IBD but also enhance the effectiveness of treatment. There is no definitive proof that modulating dysregulated bacterial populations can treat IBD effectively, but symptom alleviation can significantly improve a patient’s life. 

However, it is still unclear why only some family members who share genetic and environmental traits develop IBD. Is it a stroke of luck or a “perfect storm” of aligning factors that cause the disease? Should we blame the specific taxa found to be differentially abundant in this work for mediating the inflammatory process, or do genetic traits spark the onset and the microbiome follows? Further research on this subject remains essential for collecting additional data and delivering answers.

## 4. Materials and Methods

### 4.1. Samples

Nine CD (*n* = 7) and UC (*n* = 2) probands younger than the age of 16 were recruited from the Pediatric Department of “Attikon” General University Hospital (Table 2). All patients were newly diagnosed and drug-naïve at the time of fecal sample collection. The diagnosis was based on clinical symptoms, laboratory tests, and histological, radiological, and endoscopic findings based on the Porto Criteria recommended by The European Society for Pediatric Gastroenterology Hepatology and Nutrition (ESPGHAN). Family members of these probands were recruited and none of them had a history of IBD or other immune- and/or inflammation-related disorders. Fecal samples from all participants were obtained by Fecal Swab Collection and Preservation System (Norgen BioTek Corp, Thorold, ON, Canada) according to the manufacturer’s instructions. Fecal calprotectin was measured by an enzyme-linked immunosorbent assay (MyBioSource, Inc., San Diego, CA, USA) according to the manufacturer’s instructions. Informed consent was obtained from the parents, and methods were carried out in accordance with relevant guidelines and regulations. The research and all associated experimental protocols were performed in accordance with institutional approval from the hospital ethics committee.

### 4.2. DNA Extraction and 16S rRNA Amplicon Sequencing

Total DNA was purified from the fecal samples using the Stool DNA Isolation Kit (Norgen BioTek Corp, Thorold, ON, Canada) following the manufacturer’s instructions. Sequencing services were performed by external independent facilities (MR DNA -Molecular Research LP, Shallowater, TX, USA). Sequencing was performed at MR DNA (www.mrdnalab.com (accessed on 1 July 2023), Shallowater, TX, USA) on a MiSeq following the manufacturer’s guidelines. Sequenced reads were quality-controlled, with sequences < 150 bp and containing ambiguous base calls removed. After dereplication, the unique sequences were denoised and had chimeras removed based on the standard QIIME2 [78] pipeline using DADA2 [79], providing a denoised sequence or zOTU. Final zOTUs were taxonomically classified using BLASTn [80] (99% sequence similarity) against a curated database derived from NCBI. The final library contained samples with an average of 30,000 aligned reads. 

### 4.3. 16S rRNA Bioinformatics Analysis 

Aligned raw read counts, sample metadata, and taxonomy information files were formatted and used as input to the MicrobiomeAnalyst [81] platform for analysis. The original total of 1099 zOTUs was filtered to 384 after accounting for low read counts (<20% prevalence in all samples). Samples were normalized using total sum scaling (TSS) to avoid isolation/sequencing biases. All samples were included for the IBD versus healthy controls (HC) comparisons and sub-grouped for the per-family analyses. Alpha diversity analysis (unfiltered raw counts) was performed using the Shannon index on the zOTU level, beta diversity was calculated via NMDS and ANOSIM, and univariate differential abundance calculations were performed on the genera and species levels using DESEQ2 considering their fold change (FC) and false discovery rate adjusted *p*-values (*p.adjust*). Intrafamily microbiome similarity clustering was performed using Ward’s hierarchical clustering method. To ensure the reproducibility of results, we include in the Appendix A the necessary input files for MicrobiomeAnalyst from our data: “Appendix A”, “Appendix A”, and “Appendix A”; these files contain the raw counts, metadata, and taxonomic classification of our samples, respectively. Intersections of microbial taxa between families were created using the multiple-list intersection of the Molbiotools online platform [82].

## Figures and Tables

**Figure 1 biomedicines-11-01979-f001:**
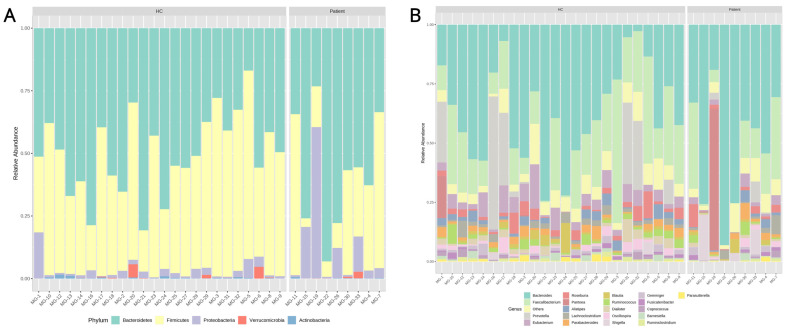
Stacked bar plots representing the bacterial relative abundance of (**A**) healthy family members and patients at the phylum level and (**B**) healthy family members and patients at the genus level.

**Figure 2 biomedicines-11-01979-f002:**
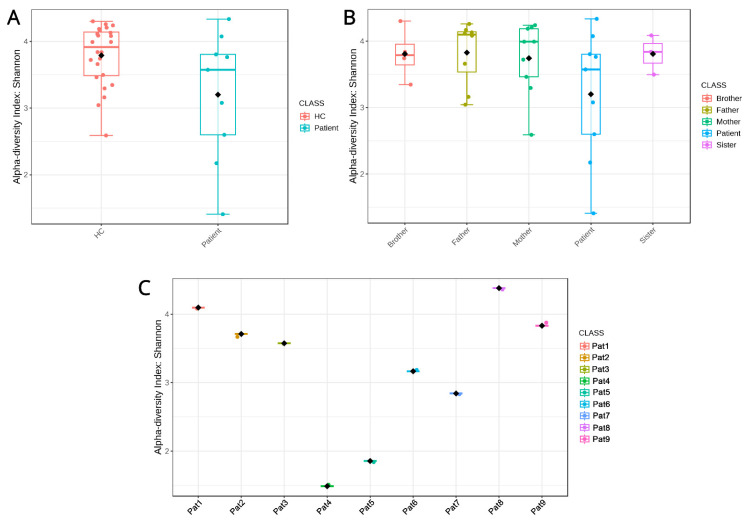
Alpha diversity indices (Shannon index) representing how biodiverse the sample groupings are. (**A**) Comparing healthy controls, in total, versus pediatric IBD patients. (**B**) Comparing healthy controls grouped by their familial relationship with the patients versus those patients. (**C**) Comparing individual patients from all families among themselves.

**Figure 3 biomedicines-11-01979-f003:**
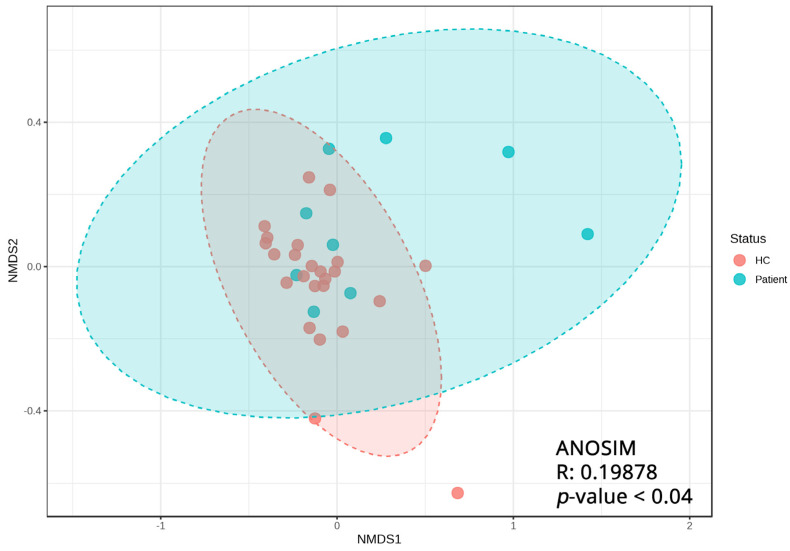
Non-metric multi-dimensional scaling (NMDS) plot depicting the qualitative and quantitative (dis)similarities between samples of healthy controls and pediatric IBD patients.

**Figure 4 biomedicines-11-01979-f004:**
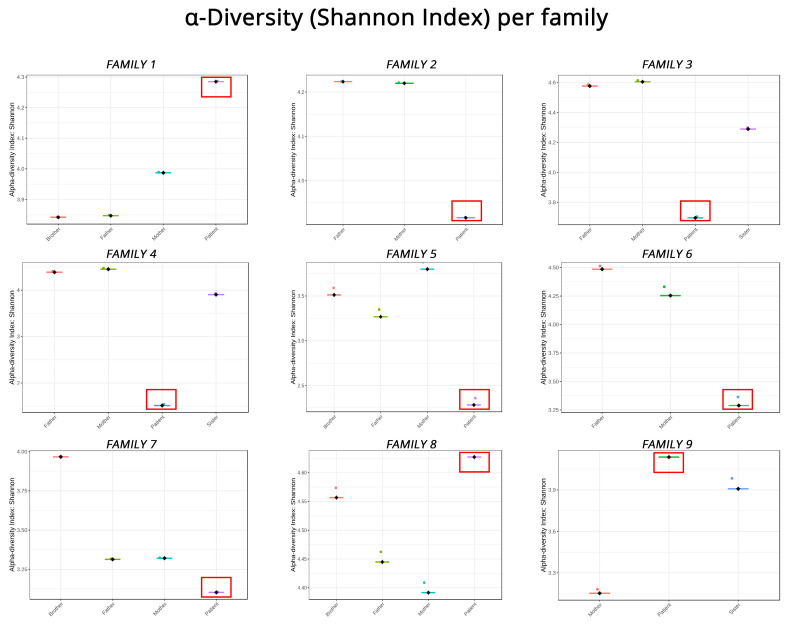
Alpha diversity indices (Shannon index) representing how biodiverse the members of each family are when compared to each other. Red boxes highlight the patients in each family.

**Figure 5 biomedicines-11-01979-f005:**
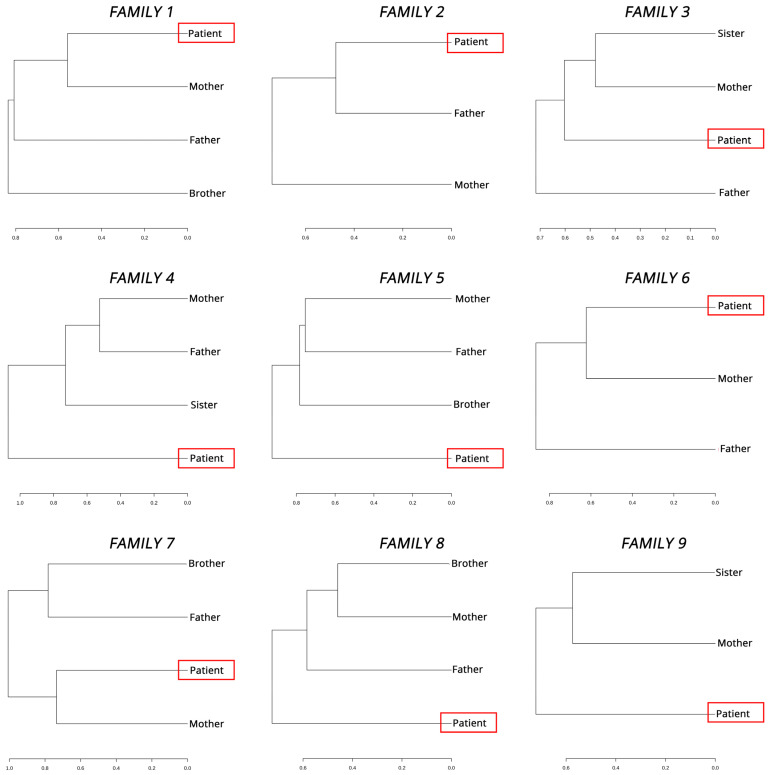
Hierarchical clustering trees for each family. Clustering family members within each family based on their microbiota composition allows us to detect similarities among them. Red boxes highlight the patients in each family.

**Figure 6 biomedicines-11-01979-f006:**
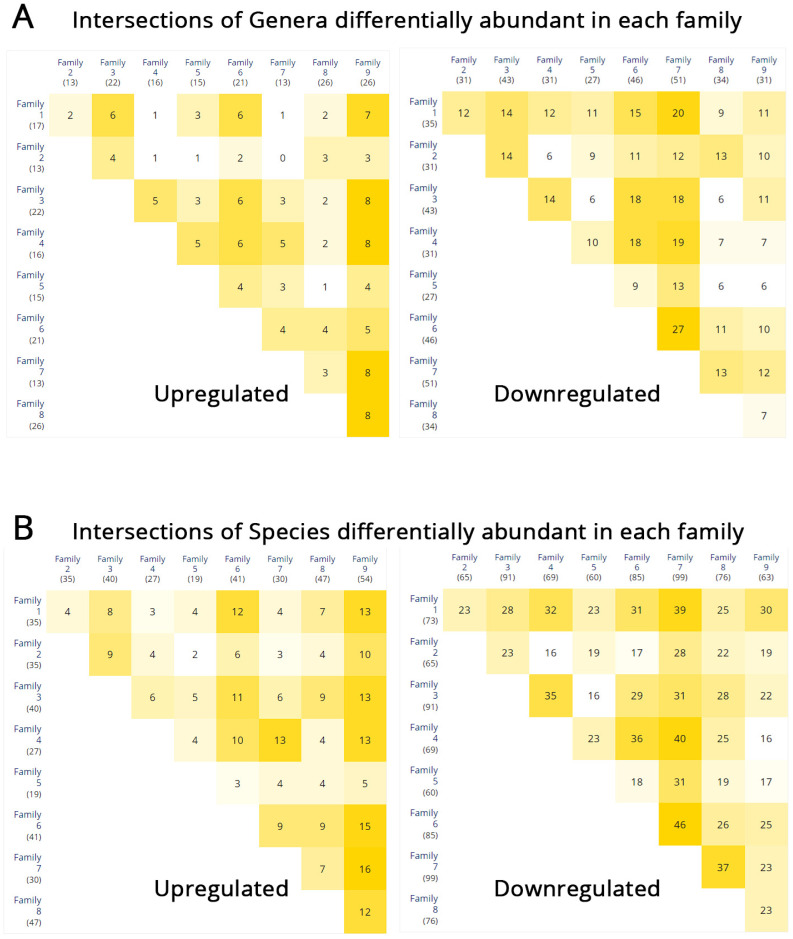
Intersections of differentially abundant taxa (FC > 2 upregulation or downregulation of abundance) among families based on intrafamilial differential abundance comparisons between patients and non-IBD members. (**A**) Intersections of microbial genera. (**B**) Intersections of microbial species. Color intensity represents higher intersection values. Numbers in each box are the number of differentially abundant taxa in each intersection.

**Table 1 biomedicines-11-01979-t001:** Top 10 over- and under-abundant species and genera in the pediatric patient group versus all healthy controls.

Genera More Abundant in IBD	Genera Less Abundant in IBD	Species More Abundant in IBD	Species Less Abundant in IBD
*Veillonella*	*Candidatus Soleaferrea*	*Veillonella_parvula*	*Ruminococcus__flavefaciens*
*Haemophilus*	*Prevotella*	*Haemophilus_parainfluenzae*	*Bacteroides__massiliensis*
*Granulicatella*	*Holdemanella*	*Erysipelatoclostridium_ramosum*	*Prevotella__copri*
*Sutterella*	*Anaerobacterium*	*Granulicatella_paradiacens*	*Bacteroides__stercoris*
*Shigella*	*Lactobacillus*	*Streptococcus_parasanguinis*	*Dialister__succinatiphilus*
*Erysipelatoclostridium*	*Holdemania*	*Bacteroides_caccae*	*Lachnoclostridium__xylanolyticum*
*Pantoea*	*Catenibacterium*	*Shigella_sonnei*	*Eubacterium__hallii*
*Streptococcus*	*Lachnospira*	*Sutterella_wadsworthensis*	*Alistipes__indistinctus*
*Butyricimonas*	*Dorea*	*Bacteroides_fragilis*	*Holdemanella__biforme*
*Alcaligenes*	*Anaerovorax*	*Bacteroides_acidofaciens*	*Anaerobacterium__chartisolvens*

**Table 2 biomedicines-11-01979-t002:** Patient demographic data.

Family Identifier	Age at Diagnosis	Gender	IBD Phenotype	Fecal Calprotectin (μg/fecal gr)	CRP ^1^ (mg/dL)	ESR ^2^ (mm/h)	PUCAI/PUCDAI ^3^	Birth Mode	Prematurity
1	13	F	CD	250	<3	1	25 (Mild)	Cesarean	Yes
2	12	M	CD	530	<3	40	30 (Moderate)	Cesarean	No
3	14	F	UC	600	8	50	20 (Mild)	Vaginal	No
4	4.5	F	CD	95	96	120	>40 (Severe)	Vaginal	No
5	16	M	CD	1660	110	65	>40 (Severe)	Vaginal	No
6	9	M	CD	1200	100	40	>40 (Severe)	Cesarean	No
7	13	M	CD	1500	40	30	37.5 (Moderate)	Cesarean	Yes
8	16	M	CD	1200	20	55	>40 (Severe)	Cesarean	No
9	13.5	M	UC	1270	<3	15	50 (Moderate)	Vaginal	No

^1^ CRP: C-reactive protein; ^2^ ESR: erythrocyte sedimentation rate; ^3^ PUCAI/PUCDAI: Pediatric Ulcerative Colitis Activity Index/Pediatric Crohn’s Disease Activity Index.

## Data Availability

Data available as Appendix A.

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
