# Peer review of "Identifying Microbiome Dynamics in Pediatric IBD: More than a Family Matter"

_biomedicines, 2023, doi:10.3390/biomedicines11071979_

Round 1

Reviewer 1 Report

The manuscript by Dovrolis et al. presents data based on a small cohort of patients, however, the findings are valuable.

The authors need to clarify several points.

1. Section 2.2.1 states that the alpha-diversity indices of patients are more or less equal (Suppl. Figure 1).

First, either Suppl Figure 1 is mislabeled and Fam should be patient, or Suppl Figure 1 is showing the alpha-diversity indices of the families.

Second, the alpha-diversity indices shown in Suppl Figure 1 do not appear more or less equal.

2. In Figure 2, the patients should be identified, so that the reader can compare Figure 2 with Figure 4.

3. The authors need to be more specific about the significance of Figure 6.

4. The Discussion states "Considering the low biodiversity exhibited by patients in all families a few things can be deduced."

It is not clear why the authors conclude that the patients have low biodiversity. Figure 2 indicates that three patients have low biodiversity, but the biodiversity of the other patients is similar to the HC. Three patients is too few on which to base a general statement about low biodiversity.

5. The Discussion states "that alpha-diversity among patients is comparable", which does not seems to agree with Suppl Figure 1.

6. The Discussion states "Additionally some of the differentially abundant taxa characterizing patients observed here might also be prominent in those children." The authors need to specify which taxa they are referring to. Perhaps a table with the "important" taxa could be added to the manuscript or supplementary data, in addition to the extensive tables in Supplementary File 1.

7. The Discussion states "The modulation of some of the taxa highlighted here has been investigated as a potential therapeutic strategy and our findings may provide more targets." Again, the authors need to be more specific about possible targets.

8. The Abstract states "We identified patterns of dysbiosis and hallmark microbial taxa among patients who shared ethnic, habitual, and dietary traits with themselves and their families. In addition, we examined the impact of the disease on specific microbial taxa and how these could serve as potential biomarkers for early detection." And, the authors list several microbial genera and with increased and decreased abundance in the patients. A paragraph in the Discussion should be added that link these two points. Again, a short Table may aid the general reader's understanding.

9. One of the conclusions stated in the abstract is that if maternal factors have a role in the establishment and modulation of the early life microbiome, this and may have implications for understanding the etiology and progression of IBD. The authors should discuss the specifics of this conclusion.

None

Author Response

The manuscript by Dovrolis et al. presents data based on a small cohort of patients, however, the findings are valuable.

Comments:

The authors need to clarify several points.

  1. Section 2.2.1 states that the alpha-diversity indices of patients are more or less equal (Suppl. Figure 1).

First, either Suppl Figure 1 is mislabeled and Fam should be patient, or Suppl Figure 1 is showing the alpha-diversity indices of the families.

Second, the alpha-diversity indices shown in Suppl Figure 1 do not appear more or less equal.

  1. In Figure 2, the patients should be identified, so that the reader can compare Figure 2 with Figure 4.

  1. The Discussion states "Considering the low biodiversity exhibited by patients in all families a few things can be deduced."

It is not clear why the authors conclude that the patients have low biodiversity. Figure 2 indicates that three patients have low biodiversity, but the biodiversity of the other patients is similar to the HC. Three patients is too few on which to base a general statement about low biodiversity.

  1. The Discussion states "that alpha-diversity among patients is comparable", which does not seems to agree with Suppl Figure 1.

Response:
We greatly appreciate your valuable comments, as they have facilitated the identification of a specific aspect in our manuscript that requires further clarification. Allow us to provide a more concise explanation.

In Figure 4, when comparing different families, it is evident that a significant proportion (6 out of 9) of patients exhibit lower a-diversity compared to their family members. Conversely, in 3 families, patients display higher a-diversity than their family members. To investigate this phenomenon, we focused solely on the a-diversity of patients within their own group (Suppl. Fig. 1). Our analysis revealed that, despite variations within the patient group, certain observations were noteworthy. For instance, Patient 9 possesses considerably higher biodiversity than their family members (as depicted in Figure 4). However, when examining the Shannon index in Suppl. Fig. 1, it becomes apparent that their biodiversity is relatively similar to Patient 2’s, who has lower biodiversity than their own family members. This highlights the complexity of the matter we sought to convey. Although the group of "Patients" exhibits lower median and mean values compared to other family groups (as illustrated in figure 2), within their respective families, patients can either display greater or lesser biodiversity.

To enhance clarity, we have revised the supplementary figure to emphasize that it represents the Shannon indices of patients, and it is now incorporated into Figure 2. We hope that this modification helps establish a better connection with Figure 4 and facilitates a clearer understanding of our findings. As mentioned previously in the Discussion, we share the Reviewer's viewpoint that the sample size could potentially restrict the generalizability of our findings. While our observations are primarily derived from the present results, they are substantiated by existing literature and our own previous research, particularly regarding the diminished biodiversity among individuals with IBD. In response to this concern, we have carefully reviewed the manuscript and implemented several revisions aimed at improving the generalizability of our conclusions. These changes were made to address the scope and limitations of our study, ensuring that our findings can be applied to a broader context. The specific revisions made throughout the manuscript to address this concern and enhance the generalizability of our conclusions are marked with “Track changes”.

 Comment:

  1. The authors need to be more specific about the significance of Figure 6.

Response:
The results have been expanded to further explain the significance of figure 6 and provide more examples of how the analysis can help us identify commonalities between patients.

The following passage was added to the manuscript: “To extend our previous analyses of microbiota differential abundance we performed the same analysis in each family comparing the pediatric patients against a pool of their family members. Additionally, we calculated the intersection of microbial genera (Figure 6a) and species (Figure 6b) with increases and decreases in their counts. This analysis enables us to identify patients who exhibit shared differentially abundant taxa. For instance, patients belonging to families 1 and 2 have only two enriched genera in common (Butyricimonas and Lachnoclostridium) but share 12 genera (Anaerosporobacter, Anaerobacterium, Prevotella, Dorea, Holdemanella, Paraprevotella, Clostridium, Ruminococcus, Lachnobacterium, Pantoea, Peptococcus, and Shigella) that demonstrate decreased population levels. Similarly, patients 3 and 6 share six enriched genera (Veillonella, Haemophilus, Shigella, Pantoea, Bifidobacterium, and Flavonifractor) and 18 genera (Phascolarctobacterium, Akkermansia, Anaerobacterium, Trigonala, Holdemanella, Intestinimonas, Spiroplasma, Alistipes, Caloramator, Clostridium, Ruminiclostridium, Sporobacter, Vallitalea, Victivallis, Holdemania, Odoribacter, Coprobacter, and Porphyromonas) displaying reductions in population size”

Comments:

  1. The Discussion states "Additionally some of the differentially abundant taxa characterizing patients observed here might also be prominent in those children." The authors need to specify which taxa they are referring to. Perhaps a table with the "important" taxa could be added to the manuscript or supplementary data, in addition to the extensive tables in Supplementary File 1.

  1. The Abstract states "We identified patterns of dysbiosis and hallmark microbial taxa among patients who shared ethnic, habitual, and dietary traits with themselves and their families. In addition, we examined the impact of the disease on specific microbial taxa and how these could serve as potential biomarkers for early detection." And, the authors list several microbial genera and with increased and decreased abundance in the patients. A paragraph in the Discussion should be added that link these two points. Again, a short Table may aid the general reader's understanding.

Response:
Supplementary file 2 was created to highlight all differentially abundant taxa of patients in each family and which of those are most common between families. In addition, Table 1 was created to show the top differentially abundant genera and species from the CD versus HC comparisons. We have also expanded that passage to reflect upon the similar findings in the literature for the 2 bacterial genera (Butyricimonas and Veillonella) which were found more abundant across patients in most families.

Comments:

  1. The Discussion states "The modulation of some of the taxa highlighted here has been investigated as a potential therapeutic strategy and our findings may provide more targets." Again, the authors need to be more specific about possible targets.

 Response:

The expanded Discussion includes recent findings and their associations with our own research, shedding light on their potential to unveil new treatment options or aid in optimizing existing ones.

 Comment:

  1. One of the conclusions stated in the abstract is that if maternal factors have a role in the establishment and modulation of the early life microbiome, this and may have implications for understanding the etiology and progression of IBD. The authors should discuss the specifics of this conclusion.

 Response:

This specific issue was argued in the Discussion. To enhance visibility, clarity and provide additional information we have expanded the section.

Reviewer 2 Report

Authors used metagenomics to analyize microbiome composition of 9 pediatric Crohn's disease patients and 24 healthy family members. Study is relevant (as pediatric CD is serious and unsolved disease of currently unclear ethiology). Study is potentially interesting, but methodology requires further consideration and clarification, and novelty / contribution over many currently published studies should be elaborated on. 

Comments:

- Per-family analysis of microbiome is interesting (as it by design corrects for many typical confounders which would be shared within families), but by design it uffers from low power due to limited sample size. Authors should also consider across-family IBD signatures for example by constructing multivariate mix models with family ID modelled as random effect

- Study is limited in size, scope, and novelty and its contribution to growing body of IBD-microbiome literature should be elaborated on in more detail in Discussion. For example, including more rigorous comparison to known results from recently published large scale IBD-microbiome would add extra value to the manuscript. 

- Pie-chart plots (Fig. 1) are difficult to compare, bar charts or equivalent would make the results more readable. 

- Methods should be elaborated on: how was denoising of RNAseq data and chimera removal performed? Which database(s) were used for classification? Please provide raw data and codes/scripts used for data processing and analysis if possible. Please specify which (if any) potential confounders were accounted for in abundance analysis (for example, Age and medication use should be examined for potential confounding effect). 

Round 2

Reviewer 2 Report

Authors have significantly improved the manuscript and addressed all my major comments. Aside from few minor formatting adjustments, I can recommend the manuscript for publication.

Minor comments:
- please ensure that resolution and style of Figures is appropriate for the final submission (Fig.1. uses different theme from the remainder of figures), also please increase the text size in Figure 1 for clarity.
- Some of figures seem to be oversized compared to the manuscript formatting, please discuss with editors and adjust as necessary.

Language through the manuscript is correct and appropriate, but there are some minor typos that should be corrected (e.g. 2.1.1. Microbiota Composition and univarifate analysis)

Author Response

We would again like to wholeheartedly express our gratitude to the reviewers for helping us improve our manuscript. We thank you for your time and efforts.

Please find below our point by point answers:

REVIEWER 2

Comments and Suggestions for Authors

Comment:
Authors have significantly improved the manuscript and addressed all my major comments. Aside from few minor formatting adjustments, I can recommend the manuscript for publication.

Response:
Again, thank you.

Minor comments:

Comments:
- please ensure that resolution and style of Figures is appropriate for the final submission (Fig.1. uses different theme from the remainder of figures), also please increase the text size in Figure 1 for clarity.

- Some of figures seem to be oversized compared to the manuscript formatting, please discuss with editors and adjust as necessary.

Response:

Figure 1 has been amended to be in line with the rest of the figures. As you can see in the provided link (https://drive.google.com/drive/folders/184fCDBZkh4wMf1bYLzBgAqP6Ttd9hnY6) our figures are high resolution and can be easily read. We will discuss the in-document formatting with the editorial staff in length during the proofing process to ensure they are all similar and appropriately placed/resized for the manuscript. 

Comments on the Quality of English Language

Comment:

Language through the manuscript is correct and appropriate, but there are some minor typos that should be corrected (e.g. 2.1.1. Microbiota Composition and univarifate analysis)

Response:

We have gone through the entire manuscript again to ensure there are no more typos or grammatical errors. Among them was the one provided by the reviewer which now reads “univariate analysis”.